# Three clusters of content-audience associations in expression of racial prejudice while consuming online television news

**Masanori Takano**[1]☯*, **Fumiaki Taka**[2]☯, **Soichiro Morishita**[1], **Tomosato Nishi**[3],
**Yuki Ogawa**[3]

**1** Akihabara Laboratory, CyberAgent, Inc., Tokyo, Japan, **2** Faculty of Human Science, Kanagawa University, Kanagawa, Japan, **3** College of Information Science and Engineering, Ritsumeikan University, Shiga, Japan

☯ These authors contributed equally to this work.
* takano_masanori@cyberagent.co.jp

**Data Availability Statement:** Our dataset is available in Supporting information.

**Funding:** F.T. and Y.O. were funded by CyberAgent, Inc. The company's employees (M.T. and S.M.)

## Abstract

It is well investigated that the expression of racial prejudice is often induced by news coverage on the internet, and the exposure to media contributes to the cultivation of long-term prejudice. However, there is a lack of information regarding the immediate effects of news delivered through television or television-like media on the expression of racial prejudice. This study provides a framework for understanding such effects by focusing on content-audience associations using the logs of an "online television" service, which provides television-like content and user experiences. With these logs, we found an association between the news-watching and comment-posting behaviors. Consequently, logs relevant to two distinct forms of racism, modern and old-fashioned racism, were extracted. Using mathematical modeling, which considers the different levels of program inducements to racist expression, personal inclinations of audiences to racism, and certainty of prediction of audience behaviors, we found three associative patterns between the news programs and audiences. The relevance of the topics covered to the basic beliefs of each form of racism was characterized into three clusters: expression as a reaction to news that is directly relevant to the basic beliefs of racism with weak inducements by non-bigots, minority abuse by distorting the meanings of news content indirectly relevant to the beliefs but with strong inducements by audiences with a strong bias, and racial toxic opinions independent of the news content by clear bigots. Our findings provide implications for inhibiting the expression of online prejudice based on the characteristics of these patterns.

## Introduction

The development and popularity of the internet have increased the importance of audiences in the trajectory of news coverage, and the current news audiences are not merely passive consumers of news content. Television news audiences can post political comments and read those posted by others on social media (second screens) during and/or after watching [1]. The

conducted this study with F.T. and Y.O. The funder provided support in the form of salaries for authors M.T. and S.M., but did not have any additional role in the study design, data collection and analysis, decision to publish, or preparation of the manuscript. The specific roles of these authors are articulated in the 'author contributions' section.

**Competing interests:** M.T. and S.M. are employees of CyberAgent, Inc. (one of the parent companies of AbemaTV). There are no patents, products in development, or marketed Cover Letter products to declare. This does not alter our adherence to PLOS ONE policies on sharing data and materials.

audiences of online news can also post and read comments on each content [2, 3]. In a remarkable number of cases, these comments contain hate speech against racial/ethnic minorities [4–8]. However, the association between news content and racial expression prejudice remains uninvestigated. In this study, we reveal the audience types that express their prejudice toward racial/ethnic minorities based on the type of news content on television-like media. We are particularly interested in how Japanese internet television viewers express racial prejudice against Koreans.

In the long-term, television news can affect the political attitude of the general audience [9]. Watching television news can reinforce their prejudice against minorities (immigrants and racial/ethnic groups) [10–14], as well as tempt them to vote intolerantly on issues such as healthcare [15–17] and immigration policies [14, 18]. This is partly owing to broadcasters' use of biased information; for example, they tend to overrepresent the crimes committed by racial minorities [10, 19]. Commercial news has stronger detrimental effects on tolerance than public news because it tends to report sensationalized tabloid topics to appeal to the general population [13].

In contrast to the long-term effects of television news, the short-term effects on the expression of racial prejudice have not been investigated yet. The latter type of news effects has been studied mainly in terms of news on the internet, rather than television owing to rich datasets that can connect each news article and reader responses (i.e., comments). Considering the topic of this study, the relationships between online news topics and the expression of reader prejudice have been investigated [4–6, 8].

The short-term effects of news broadcast on television or similar media (e.g., internet television), which many people consume at present, should also be investigated, because these news media provide experiences that differ from those with online news investigated in previous studies [20, 21]. In contrast to online news sites where users can read each news content at their own pace and when it is convenient for them, television-like media requires them to watch it at fixed pace during scheduled time. Many audiences watching the same program simultaneously, and post and read comments live. With the passage of time, the contexts in those comments are generated quickly become invisible for general audiences. In other words, comments on internet television are volatile, while other forms of online news have accumulated comments field. With these characteristics, television-like media affect the audience differently, such as imposing a time-pressure and enhancing thoughtless commenting behavior. Thus, news transmitted via television-like media can have different short-term effects on audiences in different ways from other forms of online media.

Recent audience researches have tried to capture "audience evolution [22]." That is, increasing media selectivity and interactivity due to internet technology are assumed to have changed the audience from passive observers to active participants [23, 24]. Audiences can communicate with other audiences and content providers by posting comments [22, 25]. Instead of television itself, focusing on television-like media might enable us to understand this ongoing evolution better because some of them have been integrated into the internet community more firmly.

Before describing studies of news effects on the expression of racial prejudice, we will discuss the forms of racism in today's societies. In this context, racism refers to prejudice and discrimination due to race or ethnic memberships. Prejudice, in turn, is defined as "the set of affective reactions, we have toward people as a function of their category memberships [26]." Affective reactions are based on certain beliefs on traits or behaviors of the group (i.e., stereotypes). Discrimination refers to behavioral bias and implies inappropriateness and unfairness [27]. In addition, we define hate speech "as abusive speech targeting specific group characteristics, such as ethnic origins" [28]. Hate speech has been defined in a variety of ways by various

researchers, law-makers, activists, and others. However, this simple definition (without reference to purposes, inner motives or ultimate consequences) has an advantage of enabling to automatically detect hate speeches with external information such as written comments.

It is worth noting that the research on racism since the late 20th century discussed that widely shared social norms restrain racial bias. The modern or symbolic racism theory has been developed to understand racism under such circumstances [29–31].

Racial prejudice in the pre-Civil-Right era is called old-fashioned racism, which believes in the racial inferiority of the Blacks and blatant expression of racial antipathy [30]. This type of racism has become socially unacceptable since the mid-20th century. However, according to modern racism theories, racial antipathy has transformed instead of disappearing [29–31]. Modern racism refers to racial prejudice based on the denial of the persistence of racial discrimination against the Blacks, attribution of their economic situations to their own laziness, blame for their allegedly excess demand, and belief on their privileges [31]. This form of racism is easy to accept and express, even today, owing to its appearance as mere political individualism and not racial bias [30]. Nonetheless, it has its roots in racial antipathy, similar to old-fashioned racism, and is a type of racism [31, 32].

It is worth noting that there exist other conceptualizations of today's racism besides old-fashioned and modern racism (or symbolic racism). For example, biological and cultural racism [33] or blatant and subtle racism [34] are also frequently used instead of old-fashioned and modern racism. Among them, in this article, we depend on McConahay's terminology [30] for the following two reasons: it was technically difficult to automatically detect beliefs on biological and cultural casualties from the posted comments, thus the terms that refer to these facets are misleading; and, it has advantages in consistency and convenience to adopt the same conceptualization as the previous studies on racism against Koreans [35, 36], including the availability of a well-developed dictionary (shown later).

Although the concept of modern and old-fashioned racism was determined to understand racism against African Americans, it has been demonstrated that similar structures of prejudice exist for various minorities, such as women [37] and sexual minorities [38].

Despite social norms that restrain the racial bias, there are many examples of online racism against various racial/ethnic minorities, such as African Americans [4, 6, 39], Hispanics [6], Korean residents in Japan (Zainichi Koreans) [7, 36], Indigenous Australians [40], Roma [41], and Syrian refugees [42]. Some researchers have focused on the role of high anonymity in online communities that enable users to post racial toxic comments, which would otherwise not be posted publicly by the people in modern societies [4, 5, 8, 42–47]. Other researchers have stressed that the lack of social physical cues in online communication tempts users to accord to stereotypes and group norms, including about race [46]. The omnipresence of opportunities to form communities with people who share prejudicial attitudes is another factor [46], that would facilitate echo chambers [47]. Other researchers have focused on the sophisticated styles that cloak blatant racism, such as using implicit signs of racial/ethnic minorities [4, 36, 48, 49] (e.g., gang, slum, and violent criminals), or humorous transformation [40]. Modern racism is another way to express racial antipathy publicly. However, the expression of old-fashioned racism was as frequent as modern racism for Koreans (above 10% both) considering the tweets posted in Japanese that directly refer to them, which is the main objective of this study [36].

Under such circumstances, the associations between the topics of online news articles and racial bias have been investigated. For example, toxic comments, including racial prejudice, tend to be posted on the news related to politics, healthcare, religions, social issues, and celebrities [5–7, 44, 50, 51]. Racialized topics invoke inflammatory responses more frequently than non-racialized topics [44]. In addition, although most news articles are not about racial issues,

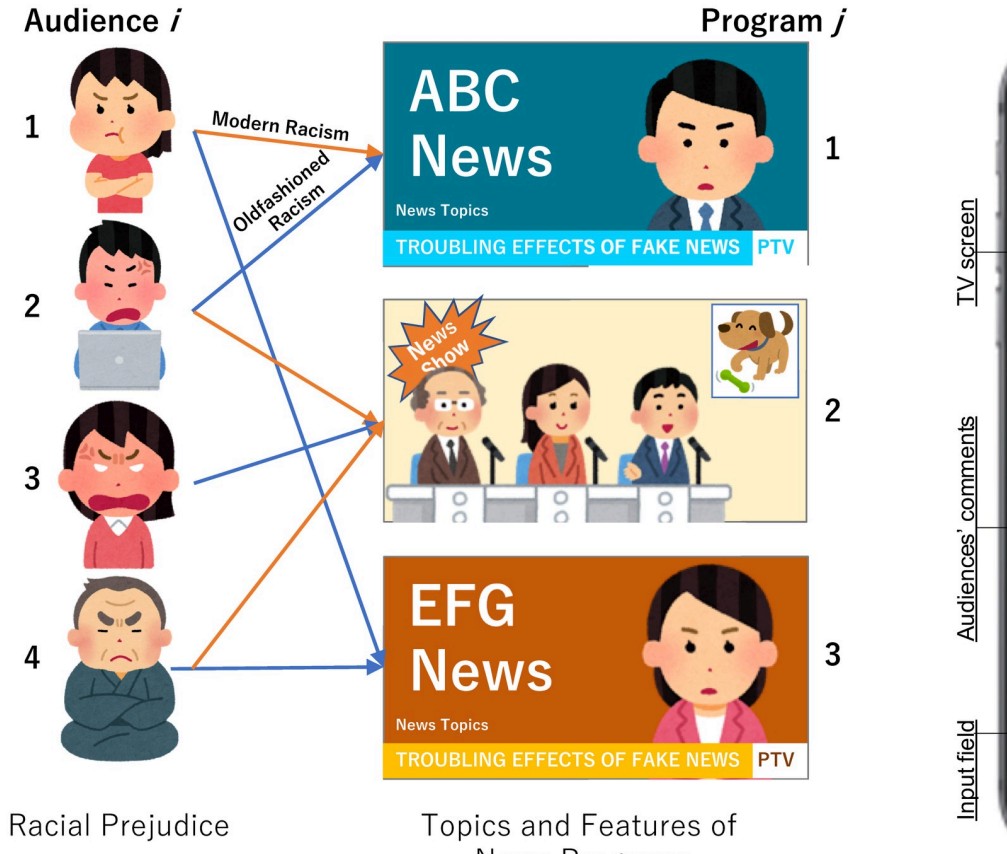

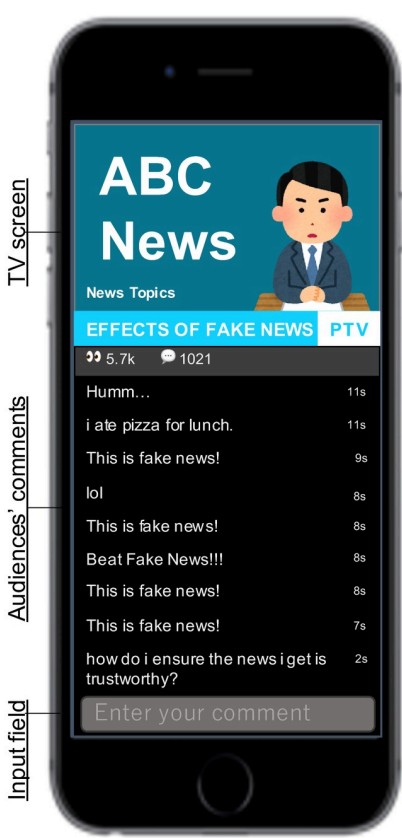

a) Interactions between audiences and news contents  b) A mock screen of ABEMA

**Fig 1. Online television and audiences.** (a) Racial prejudice expression is a result of the interaction between news topics and two types of racial prejudice. (b) Audiences can share their opinions as comments while watching news.

news readers post racist comments irrespective of their clear irrelevance [6]. For example, some readers baselessly label the criminals featured in news articles as racial/ethnic minorities and post racist comments [6], whereas some post racist comments on news topics are irrelevant to racial issues, such as weather or animal news [5, 6].

Traditionally, media exposure is one of the major pathways through which modern people learn racism [30]. As noted above, recently, the audience responses are an additional source of information for other audiences [1–3]. Thus, understanding the association between news content, users, and their direct responses (Fig 1a) will help interpret today's racism issues. However, little is known about the direct effects of television news exposure on the audience response in compared to online news websites, as noted above, owing to a lack of dense datasets that can connect those factors.

To compensate for the lack of datasets, we use the logs of news-watching behaviors and comment activities on a Japanese online television provider, "ABEMA" (https://abema.tv/). The "online TV" category can include several media types, such as YouTube, which allows general users to upload their own content, or Netflix, which enables users to watch the stocked official videos according to their convenience. Among them, the news channel of ABEMA only provides official programs created by one of the largest Japanese broadcasting companies,

"TV Asahi," and allows users to watch each program at a certain time. Thus, its audience experiences are identical to those with programs delivered via radio waves by traditional broadcasters.

Besides the resemblance to traditional television, ABEMA has comment sections (50 character limit) that can be user by users to share their comments with others in real-time (Fig 1b). In other words, ABEMA, as a single medium, has both a television screen and a second screen. User comment logs contain information regarding the programs, users, and time of posting comments.

The combination of similarity of ABEMA with television and the presence of detailed logs of audience behavior (watched contents, watching duration, and comment posting) as an internet media significantly benefits the research on media communication [52, 53]. We exploit this to investigate the immediate effects of news exposure.

Among the other Japanese websites that allow users to post comments [7, 36], ABEMA has numerous racist comments. Although ABEMA has devoted large efforts to decrease such toxic comments (ABEMA has been striving to remove these toxic comments via natural language processing), it has not been able to prevent them completely. In this study, we use all comment data including removed comments. The rampancy of racist comments partly attributes to the high anonymity in ABEMA. The users, including those who post comments, keep their IDs/usernames invisible to other audiences (see ABEMA user interface in Fig 1b). Therefore, the ABEMA dataset provides an integrated picture of television news exposure and the expression of racial prejudice by audiences as a response to news content, which is observed in anonymous social media.

The benefit of understanding the expression of racism by people toward specific content is not limited to television and television-like media. In other words, this study can benefit ordinary online news providers (e.g., text-based news sites). Racial hate speech on news platforms does not only offend the targeted people, but also damages the platform environment by reinforcing bystander prejudice and discrimination, thereby exposing these platforms to racism [8, 54, 55].

However, the efficacy of direct restrictions, such as filtering or removing racist comments and banning racist accounts, is limited. Additionally, linguistic filtering approaches can accidentally impose the biased treatment of discriminated minorities [56]. Banning racist accounts has a limited effect on removing racist communities owing to their resilience and adaptivity [57]. Therefore, news media must confront online racism with a better understanding of the relationships between news content and expression of racial prejudice by audiences.

In this study, we focus on the Korean among minorities or outgroups as targets of racial/ ethnic prejudice. It is well documented that Zainichi Koreans are one of the main targets of serious discrimination in Japanese society [36, 58–61]. Although Zainichi Koreans account for a small percentage of the Japanese population (0.4%) and the media coverage about them was relatively rare, hate speech against them has significantly increased in the last two decades. Koreans, as an ethnic group, irrespective of whether they live in Japan, are a target of prejudice in Japan [62]. The antipathy toward Koreans by the Japanese can be partly attributed to extreme historical and territorial issues between the two Korean states and Japan [36, 58, 61, 62].

Racialization of Asian people can be traced back to the Meiji period (A.D.1868–1921) in Japan [63]. To maintain collective self-esteem while confronting well-developed western cultures, the Japanese and the government required myths that Japanese ethnicity is superior to other Asian people. During the invasion and colonial rule over Asia, the notion of racial hierarchy, which put the Japanese on the top became widely accepted. After the end of World War II, Japan waived dominance abroad, but derogation to Asian people continued [63]. In 1959,

Japanese students sample rated Koreans most unfavorable among 12 race/ethnic groups [64]. Recently, some factors aggravate racial prejudice against Koreans by Japanese people. For example, the popularization of the internet and anonymous cultures [36], the spread of historical revisionism, which affirms the Japanese Empire and denies the existence or responsibility of Japanese' war crimes [36]. Additionally, the undergoing geopolitical conflicts between Japan and South Korea and between Japan and North Korea also contribute to the racial tension [62].

In summary, we analyze the association between news content, audiences, and expression of racial prejudice against Koreans, especially modern and old-fashioned racism, using logs of ABEMA.

## Results

### Summary of findings

We divided the news logs into 5 minutes grids and observed four clusters for each type of racism: non-expression, ambiguous, evocative, and non-evocative clusters (Table 1). These clusters were characterized by inducements of news grids, individual inclinations of the average audience, certainty of prediction of user behavior from these two, and relevance of racism and covered contents.

In the non-expression clusters, the target racism was not expressed on each grid. These grids were included in the dataset because the other type of racism was expressed on them.

In ambiguous clusters, the predictions regarding of audiences posting racist comments considering their inclination, as well as the inducements of news grids, were highly ambiguous, and involved users were not considerably bigoted on average. Their comments seemed to be reactions to news contents, which have relatively direct connections to the basic beliefs of modern/old-fashioned racism (e.g., crime news for old-fashioned racism), thereby resulting in racist connotations.

In the evocative clusters, the predictions from individual inclinations and inducements of news grids had strong certainty. The grids included in this cluster were highly evocative of hate speech. However, the covered content seemed to be indirectly relevant to racial issues and involved users with a strong inclination to racism. The bigots posted racist comments as a reaction to news contents based on their opinion toward racism; however, they seem illogical to the common people.

In the non-evocative cluster, user inclinations predicted user behaviors with certainty. The included content was barely relevant to racial issues and was not evocative of hate speech. Biased users posted toxic comments, such as propaganda of an extreme-right group and claims of inferiority of Koreans, irrespective of the covered content. Thus, unconditional hate speech characterized this cluster.

**Table 1. Summary of findings.** The relationships between news grid clusters and patterns of racial prejudice expressions. Derivations of cluster names were as follows. *Ambiguous*: Which audiences express racial prejudice on what grids were difficult to predict from the characteristics of the audiences and news; *Evocative*: News grids evoked audiences to express racial prejudice due to some characteristics of the contents; and *Non-evocative*: News grids itself did not evoke audiences to express racial prejudice, but some audiences did so independently of news contents.

| News grid cluster | Relation with the beliefs of racism | Expression pattern |
|---|---|---|
| Ambiguous | Direct relevance | Reactive |
| Evocative | Indirect relevance | Reactive with leap in logic |
| Non-evocative | Irrelevance | Unconditional |
| Non-expression | Irrelevance | – |

## Clusters of news grids

We evaluated news grids statistically to separate the factors related to the audience and news grids because the expression of racial prejudice depends on both [5, 6, 44, 65]. For this separation, we considered that a prejudice expression $y_{ij} \in \{0, 1\}$ on news grid $j$ depends on inducements $b_j \in (-\infty, \infty)$ of the news grids.

In addition, we adopted $\theta_i \in (-\infty, \infty)$ (audience $i$'s inclination to express racism) and $a_j \in [0, \infty)$ (grid $j$'s certainty of prediction based on $b_j$ and $\theta_i$; $a_j = 0$ means racism was expressed irrespective of audience inclinations and news grid inducements). $a$ and $b$ are the news grid factors, and $\theta$ is the audience factor. Based on this consideration, we used the following statistical model [66].

$$y_{ij} \sim \text{Bernoulli}(p_{ij}) \tag{1}$$

$$\text{logit}(p_{ij}) = a_j(\theta_i + b_j + \beta), \tag{2}$$

where $\beta$ is an intercept, and Bernoulli($p$) is a Bernoulli distribution with probability $p$. We estimated $\theta_i, a_j, b_j, \beta$ for each type of racism via the Markov chain Monte Carlo (MCMC) [67]. Two types of racism were distinguished by superscripts, i.e., $\theta_i^m, a_j^m, b_j^m, \beta^m$ (modern racism) and $\theta_i^o, a_j^o, b_j^o, \beta^o$ (old-fashioned racism). We described the mean of the posterior distribution for each parameter using a hat, e.g., $\hat{\theta}_i^m$.

This model was found to fit our dataset well, with pseudo R-squared values of 0.578 [0.573, 0.583] (McFadden) and 0.931 [0.930, 0.932] (Estrella) for modern racism, and 0.387 [0.383, 0.391] (McFadden) and 0.931 [0.930, 0.932] (Estrella) for old-fashioned racism. The front of the square brackets is the median of the sampling distribution. The values in the square brackets indicate credible interval 95%.

The news grids were divided into four clusters for each racism (Fig 2; see the Clustering News Grids section in Materials and Methods) based on the presence or absence of expression of the target racism and the estimated parameters $\hat{a}_j$ and $\hat{b}_j$ (news grid factors). These clusters were labeled as *non-expression*, *ambiguous*, *evocative*, and *non-evocative*. The grids that did not contain the target racism were classified into non-expression cluster, and the remaining grids were divided into the following three clusters:

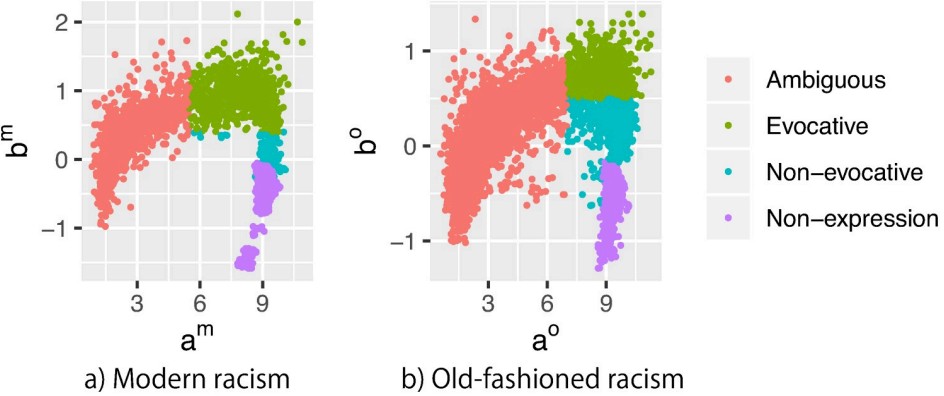

a) Modern racism b) Old-fashioned racism

**Fig 2.** Scatter plots of $\hat{a}_j$ and $\hat{b}_j$ for modern racism (a) and old-fashioned racism (b). We classified grids as three clusters using threshold lines shown in S1 Fig.

Ambiguous cluster was adopted in grids with smaller $a$ than those in other clusters. Their $b$ were distributed over a wide range, and $a$ exponentially increased with $b$. Thus, when a grid inducement $b$ was low, the expression on prejudice by audience $i$ was ambiguous irrespective of their personal inclination to racism $\theta_i$ because $a$ was also small. In contrast, when a grid had strong inducement, predictions regarding the expression of prejudice had more certainty as $a$ was relatively high. Audiences with strong inclination to racism frequently posted racist comments on grids with strong inducements. The ambiguous cluster adjoined the evocative cluster (shown next) at this end.

Evocative and non-evocative clusters were adopted in the remaining grids with larger $a$. In these clusters, the expression of racial prejudice by audience $i$ was efficiently predicted via $b$ and $\theta_i$. The definitive difference between evocative and non-evocative clusters lies in the values of $b$ for their grids. The evocative cluster was adopted in grids with high $b$, and the non-evocative cluster was adopted in those with small $b$.

S1 Table lists the frequencies and ratios of audiences, news grids, and comments by the four clusters for modern/old-fashioned racism.

Each grid was assigned to a specific cluster, as noted above, and each comment was posted on a specific grid. Thus, each grid and comment were recorded only once in S1 Table. In contrast, audiences who posted comments on multiple grids belonging to different clusters were counted more than once. However, the number of those who contributed to multiple clusters was relatively small; for modern and old-fashioned racism, only 159 (4.78%) and 335 (9.86%) of the total audience were recorded, respectively.

The ambiguous clusters were more frequent than the evocative and non-evocative clusters for both modern and old-fashioned racism. In other words, many audiences and grids were involved in the former clusters. This tendency was more conspicuous for old-fashioned racism than modern racism.

Non-evocative clusters, on the other hand, drew only a small portion of the audience. The grid/audience and comment/audience ratios were larger for the non-evocative cluster than for the ambiguous and evocative clusters. In other words, a small amount of audience posted many comments on many grids on average. This tendency was common for modern and old-fashioned racism but was more extreme for the former than the latter.

The ratio of comments that expressed modern or old-fashioned racism compared to the total number of comments that referred to Koreans was smaller than those observed on Twitter. According to a previous study on Twitter, which used a similar dictionary as this study, tweets expressing racism accounted for 12.2% and 10.8% of all tweets referring to Koreans for modern and old-fashioned racism, respectively [36]. However, in this dataset, the expression of racism accounted for 4.3% and 8.5% of the comments referring to Koreans for modern and old-fashioned racism, respectively.

In addition, comments expressing old-fashioned racism were more prevalent than those expressing modern racism, in contrast to the previous study [36]. Therefore, the intercept of old-fashioned racism, $\beta^o$ (−1.854 [−1.907, −1.800]), was larger than that of modern racism, $\beta^m$ (−2.834 [−2.902, −2.768]).

## News topic categories of three clusters

Table 2 lists the news topic categories that characterize each cluster for modern and old-fashioned racism.

**Ambiguous cluster.** The news grids of ambiguous clusters tended to report the issues directly relevant to the basic beliefs of modern/old-fashioned racism. For modern racism, the characteristic topic categories included political news, international news, and news on social

**Table 2. Frequencies and pointwise mutual information (PMI) of news topic categories by clusters for modern and old-fashioned racism.** The category names in the Topic Category columns show the characterization of each cluster (PMI > 0). Category names were sorted in descending order by PMI. PMI is defined as follows: $PMI_k(c)$ = $\log_2 P(c|k) - \log_2 P(c)$, where $c$ is a topic category and $k$ is a grid cluster. $PMI_k(c) > 0$ denotes the category $c$ characteristic under the given condition cluster $k$. The frequency of topic category $c$ is the ratio of the category in each cluster ($P(c|k)$). International news was the most frequent category among all clusters and both types of racism because we classified the news grids related to South Korea and North Korea as international news. This news was not about Zainichi Koreans (i.e., Korean residents in Japan), but the expression of modern racism, which is about Zainichi Koreans in specific, was observed.

| Cluster | Modern racism | | | Old-fashioned racism | | |
|---|---|---|---|---|---|---|
| | Topic Category | PMI | Frequency | Topic Category | PMI | Frequency |
| Ambiguous | Social issues | 0.512 | 0.200 | Incident | 0.441 | 0.053 |
| | Political | 0.392 | 0.110 | Crime | 0.242 | 0.154 |
| | International | 0.210 | 0.250 | Political scandal | 0.028 | 0.088 |
| | Nuclear accident | 0.097 | 0.010 | Lifestyle | 0.015 | 0.057 |
| | Disaster | 0.005 | 0.050 | | | |
| Evocative | Nuclear accident | 0.557 | 0.010 | Lifestyle | 0.371 | 0.073 |
| | Social issues | 0.403 | 0.190 | Disaster | 0.273 | 0.059 |
| | Disaster | 0.325 | 0.060 | Entertainment | 0.213 | 0.040 |
| | Political | 0.270 | 0.100 | Crime | 0.191 | 0.149 |
| | Weather | 0.263 | 0.040 | Sports | 0.097 | 0.035 |
| | Sports | 0.149 | 0.040 | Celebrity/sports scandal | 0.060 | 0.093 |
| | | | | Weather | 0.055 | 0.033 |
| Non-evocative | Celebrity/sports scandal | 1.042 | 0.180 | Nuclear accident | 0.602 | 0.014 |
| | Entertainment | 0.617 | 0.050 | Sports | 0.181 | 0.037 |
| | Political scandal | 0.158 | 0.100 | Entertainment | 0.105 | 0.037 |
| | International | 0.117 | 0.230 | Political scandal | 0.090 | 0.092 |
| | | | | Weather | 0.067 | 0.033 |
| | | | | International | 0.036 | 0.220 |
| | | | | Lifestyle | 0.034 | 0.058 |
| | | | | Crime | 0.026 | 0.133 |
| | | | | Celebrity/sports scandal | 0.020 | 0.090 |
| | | | | Political | 0.008 | 0.087 |

issues (i.e., hard news topics [68]). This news rarely demonstrated a direct reference to Koreans, excluding international news.

Modern racism was often expressed on news related to political or social issues, involving domestic vulnerable people (e.g., welfare recipient, racial minorities), and international news, which included diplomatic issues between Japan and South Korea. For example, some audiences claimed to terminate the alleged privilege of Zainichi Koreans (e.g., "easiness" of welfare receipt) on social issues news, and the same or other audiences expressed that such type of "privilege" would be terminated soon owing to the then-worsening relationship between Japan and South Korea. Old-fashioned racism was observed mainly on incident and crime news. Some audiences baselessly labeled perpetrators or criminals as Zainichi Koreans and claimed that Koreans were inferior in terms of morals, abilities, and appearances. Social welfare and criminality/inferiority are the major concerns in modern and old-fashioned racism [30], detecting the direct relevance between news content and racism could easily.

**Evocative cluster.** In evocative clusters, the relevance of news topics to the basic beliefs of target racism was not as clear as that in ambiguous clusters.

For modern racism, the characteristic topic categories included nuclear accidents (Fukushima nuclear power plants, in specific), along with disaster, political, weather, and social issues. Some audiences claimed that Zainichi Koreans are cheating tax-system on news discussing the countermeasures to natural disasters, and other or the same audiences exhibited the same

behavior on social issues news irrelevant to Koreans or the alleged privilege of Zainichi Koreans.

For old-fashioned racism, the news grids in this cluster treated familiar types of news (e.g., lifestyle and entertainment news), besides disaster and crime coverage. The news grids also lacked clear relevance to Koreans or the basic beliefs of old-fashioned racism. On such news grids, some audiences baselessly claimed that cast members of the programs are Koreans disguising themselves as Japanese and hurled abuses on them; the same or other audiences followed these comments to claim the inferiority of Koreans in terms of morals, abilities, or appearances.

**Non-evocative cluster.** In non-evocative clusters, the characteristic news topic categories were seemingly irrelevant to Koreans or the basic beliefs of target racism. For example, some audiences expressed modern racism on celebrity/sports scandal, entertainment, or political scandal news. The same or other ones expressed old-fashioned racism on news related to nuclear accidents and sports. It is evident that these categories are typically included in soft news [68]. A small number of audiences chanted the propaganda of a right-wing extremist group and denigration against Koreans in fixed phrases, irrespective of the news content. For example, the chants included claims on the termination of social welfare for Zainichi Koreans for modern racism and those regarding the inferiority of Koreans for old-fashioned racism.

## Audience and the expression patterns

Fig 3 shows the distribution of mean audience inclination $\hat{\theta}_i$ by mean grid inducements watched by audience $i$, i.e., $b'_i$ ($b'_i = \frac{1}{|J_i|} \sum_{j \in J_i} \hat{b}_j$, where $J_i$ is a set of grids audience $i$ watched), wherein the clusters are indicated with different colors, except for the non-expression clusters.

The average $\hat{\theta}^m$ and $\hat{\theta}^o$ were the largest for the non-evocative clusters, followed by the evocative clusters. The ambiguous clusters exhibited the smallest $\hat{\theta}$. In other words, audiences who posted racist comments on these clusters had stronger personal inclination toward racism on average in this order.

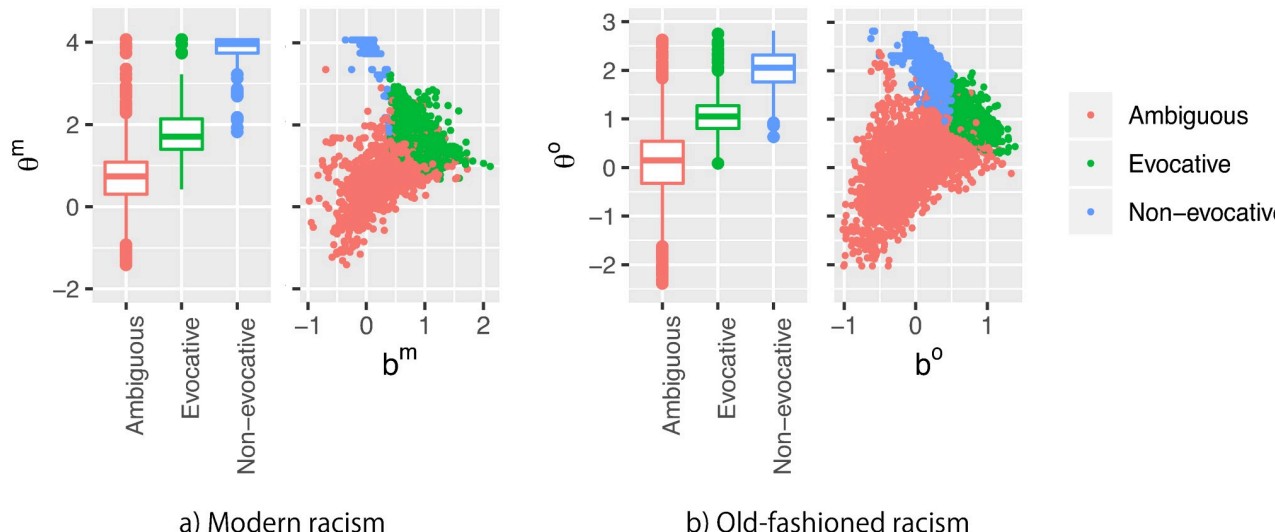

a) Modern racism b) Old-fashioned racism

**Fig 3. Distributions of mean $\hat{\theta}$ about each grid cluster.** Note that audiences who expressed racism in both clusters are plotted as the points of both clusters.

Furthermore, $\hat{\theta}$ and $b'$ correlated in different manners among the clusters. In the evocative and the non-evocative clusters, the two variables correlated negatively; the correlation coefficients were observed to be −0.223 [−0.301, −0.142] for the evocative and −0.131 [−0.208, −0.039] for the non-evocative clusters for modern racism, and −0.179 [−0.259, −0.095] for the evocative and −0.334 [−0.405, −0.255] for the non-evocative clusters for old-fashioned racism. In other words, when situational inducements were relatively weak, audiences with a stronger bias posted racist comments.

In contrast, $\hat{\theta}$ and $b'$ correlated positively in the ambiguous clusters; the correlation coefficients were observed to be 0.225 [0.183, 0.263] and 0.244 [0.216, 0.269] for modern and old-fashioned racism, respectively. Recall that $a$ was a function of $b$ in ambiguous clusters. When $b$ was small, both $\theta$ and $a$ were small. In other words, people without a strong bias were also as often involved as those with a strong bias, and the predictions based on individual inclinations exhibited less certainty when situational inducements were weak.

## Relationships between modern and old-fashioned racism

Although previous studies indicated moderate to strong positive correlations between the strength of modern and old-fashioned racism measured using questionnaires [30, 31, 36], in this study, the two types of racism were not frequently co-expressed. The audiences who expressed both types of racism accounted for 16.8% of the total audience; additionally, the news grids and comments that included both types of racism accounted for 9.5% of and 2.2% of the total number of grids, respectively. Therefore, the correlation between stimulus inducements $b^m$ and $b^o$ was almost zero (0.000 [−0.051, 0.054]). Similar observations were made for the correlation between $a^m$ and $a^o$, 0.000 [−0.049, 0.051]. Correspondingly, the co-occurrence frequencies between modern/old-fashioned racism clusters (S3 Table) demonstrated an exclusive relationship (Cohen's $\kappa$ = −0.332). For example, ambiguous cluster news grids, where audiences expressed modern racism, did not tend to include old-fashioned racism (non-expression clusters).

However, the independence of the two forms of racism disappeared when the inferred individual inclination $\theta$ was considered; $\theta^m$ and $\theta^o$ correlated positively and more than moderately (0.650 [0.611, 0.686]).

## Discussion

We found three common patterns of association between the news content and audiences on the expression of racial prejudice across modern and old-fashioned racism (Table 1). The results showed who express their prejudice on what contents and by what manners. In other words, we found three patterns of audiences engagements [22] to television-like media in the aspect of expressing prejudice.

First, there were clusters of news programs whose topics were directly relevant to the basic beliefs of either form of racism (e.g., social welfare, crimes). Regarding these programs, the indicator of audience inclination to racism, $\theta$, and situational inducements, $b$, were not good predictors of racist expression (i.e., low $a$). Thus, these clusters were called ambiguous clusters.

On these programs, audiences who were not significantly bigoted posted racially loaded comments probably as a reaction to contextual cues related to Koreans. The covered topics did not directly mention Koreans, and a large percentage of audiences did not refer to them. Although, some audiences mentioned Koreans on these programs based on the relevance of topics.

In addition, for news programs that had relatively stronger inducements in these clusters, the audience inclinations and certainty of prediction had higher values. In other words, when

the covered content had a stronger connotation of racial issues, people with stronger inclinations reacted more frequently.

It is worth noting that the largest ratios of audiences and comments included in this dataset belonged to the ambiguous clusters for both types of racism. Hate speech in comments on online streaming is likely to be delivered by people with normal behaviors who occasionally discriminate against minorities than those with a strong racial bias. It is crucial to reduce occasional discrimination by these large segments of audiences.

Second, there were clusters of news programs whose topics have indirect relevance to the basic beliefs of racism or Koreans; however, they had strong inducements of racist expression (high $b$). These clusters were thus called evocative clusters. Audiences belonging to these clusters tended to have strong personal inclinations (high $\theta$). In addition, the programs had high values of $a$; thus, audience inclinations and program inducements can help predict the audiences that posted racist comments on specific programs with certainty.

Although ordinary people consider the reported issues as irrelevant to Koreans, audiences with strong inclinations to racism ignore these gaps and connected the both. Furthermore, there were negative correlations between program inducements and user inclinations to racism. The programs in these clusters selectively affected audiences with stronger inclinations to racism, even when they had relatively weaker inducements. These findings indicate that there are some typical patterns of leap in logic, which the bigots follow to connect the covered issues and Koreans and deliver hate speech. In other words, some contextual cues, which do not affect ordinary people, trigger racist expressions by the bigots.

Third, there were clusters of programs whose topics were irrelevant to either form of racism; nevertheless, racist comments were posted on them. High values of $a$ along with low values of $b$ and the highest values of $\theta$ imply that these programs cast little inducements over the audiences in general (i.e., non-evocative clusters), and only those who have strong personal inclinations to racism posted racist comments. The negative correlation between the values of $b$ and $\theta$ for programs and audiences, respectively, further demonstrates that when the programs are less relevant to racial issues, the engaging audiences have a stronger bias.

Most of comments posted on these programs were propagation of an extremist right group's claiming existence of Zainichi Koreans' privilege for modern racism, and claims on the inferiority of ethnic Koreans in terms of morals, abilities, and appearances for old-fashioned racism. Some fixed sets of sentences or phrases for racist expression was observed. Thus, it can be concluded that the extreme bigots expressed racism without considering the news content. Some of them are organized activists from xenophobics.

It is worth noting that the number of audiences was the smallest for this cluster among the three clusters that were adopted to express each type of racism, and those audiences posted more racist comments per person. Making interventions in these small segments help reduce the amount of racism with lesser cost.

This study was conducted with a dataset of a specific type of media, i.e., online television. The findings of the three distinct clusters of association between audiences and programs, along with the concept of two types of racism are consistent with those of previous studies and provide novel insights on the phenomena of online hate speech.

Previous studies related to comments on online newspapers found that some reported topics, such as politics, healthcare, crimes, and religions, induce racist expression more strongly than other topics [5, 6, 44, 50, 51]. For example, reports on crimes tempt the audiences to label criminals as members of racial minorities baselessly [6], and other topics tempt them to post toxic comments in the form of social or political discourse [5, 6, 44, 50, 51]. The difference between these comments can be elucidated under the notion of old-fashioned and modern

racism. Another study followed the same concept to investigate racist expression among Japanese tweets, and found that both types of racism in many tweets [36].

Furthermore, the relevance of the reported topics to the aforementioned racial issues can be easily identified [6]. Thus, such types of news coverage correspond to ambiguous clusters in a previous study. Our findings mean that, like the ambiguous clusters in this study, racist comments are posted by rather normal audiences instead of bigots on topics directly relevant to the basic beliefs of racism.

It is worth noting that although the reported topics were relevant to the basic beliefs of racism, the news coverage lacked direct reference to racial minorities. The lack of connection between the reported topics and racial minorities needs to be compensated by audiences. Thus, this ease of spontaneous compensation is an important parameter in the largest part of online hate speech. A study [36] found that the outgroup, which the Japanese Twitter users frequently refer to along with themselves was the Korean, instead of Americans, despite the strong connection between Japan and the USA in political, economic, and military domains. They are "obsessed" with Koreans and connect them to any unfavorable outcomes. This widely shared tendency to easily connect them should be addressed.

Another finding of previous studies stated that some audiences post toxic comments on news coverages that are irrelevant to racial issues [5, 6]. Such news includes two types of topics, according to this study. One lacks direct relevance but has an indirect relevance to racial issues, and only strongly prejudiced audiences associate the two. The other lacks relevance to racial issues, and true bigots post racist comments on such issues, irrespective of the news content. Distinguishing these two clusters helps address the rampancy in hate speech. For example, the audiences included in the latter cluster post many comments that never enrich the discourse on news content; they only exploit the news platforms. Thus, they should be treated differently from other audiences.

This study also found that old-fashioned racism was expressed two times as frequently as modern racism. In contrast, a previous study [36] that almost used the same dictionary as this study to analyze Japanese tweets found that old-fashioned racism against Koreans was less than modern racism. The total ratio of expression of racism was smaller for the comments on ABEMA than that observed in tweets, as stated in previous study [36], which can be attributed to the character limit in ABEMA (50 characters in ABEMA v.s. 140 in Twitter) that allow users to address fewer topics per comment. Thus, the relative prevalence of old-fashioned racism should be explained in comparison to modern racism.

Besides Taka [36], other studies [40, 44] also found that people who express racial prejudice tend to pick their words carefully to avoid blatant racism. The existence of such aversion toward blatant racism corresponds to the claims of the modern or symbolic racism theorist, which states that people in the post-Civil Rights Movement era recognize that blatant racism will be rejected by society; therefore, they avoid blatant racism and express antipathy in more subtle forms instead [29–31].

Why is this not observed in the comments on ABEMA? A possible explanation is the real-time feature of ABEMA, where all audiences of each program watch at the same time. Thus, it is important to make timely comments to make the programs meaningful for other audiences. It has been argued that when people cannot exploit enough mental resources, they fail to suppress their "genuine" prejudice, and express them [69]. In the case of ABEMA, a time-constraint results in insufficient usage of mental resources.

Another possible reason is the high anonymity on ABEMA. It has been demonstrated that people tend to post toxic comments on news websites when they have the opinion to be anonymous [5, 44]. The same is true in ABEMA owing to the lack of any user identifier that can be viewed by other audiences. Twitter and other social media platforms have various types of

identifiers, although some are pseudonyms, which suppresses blatant racism to a certain extent.

The above reasons depend on the characteristics of ABEMA as media. However, the relative prevalence of old-fashioned racism can be explained considering the delivered content. The largest divergence of the frequency between the expression of the two forms of racism was observed in the ambiguous clusters. Thus, it is possible that the amount of coverage for each topic, frames selected each news, and audiences of each program affected the probabilities of expression of each type of racism as a reaction to the viewed content.

The daily conversations among families or friends are important pathways that help people in developing prejudice [70]. However, it is difficult to investigate situations in which people watch television and interact with their families or friends owing to a lack of accurate behavioral logs. This study provides implications for this issue as well. Among the three clusters, propaganda independent of news contents (i.e., comments on non-evocative clutters), which are directed to unspecified multitude of audiences, was irrelevant to these close relationships; thus, it is irrelevant to this context. However, the other two types of expression apply to family or friends' settings. Television audiences transmit their prejudiced attitude by teaching others how to connect news topics irrelevant to racial issues, as in the evocative clusters of this study. In other cases, people express prejudice as a normal reaction to news coverage, as in the non-evocative clusters of this study; such repeated access to biased beliefs makes prejudice easily accessible. Although these are just implications, these hypotheses are worth investigating in the future.

This study also provided new insights on the modern or symbolic racism study. Previous studies using questionnaire survey methods have demonstrated moderate to a high correlation between modern and old-fashioned racism [30, 31, 36], which is the basis of the claim that there are common underlying factors, such as racial antipathy, between the two forms of racism [31]. In this study, only a small percentage of comments, audiences, and news grids were associated with the expression of both forms of racism. In other words, the two forms of racism have rarely co-occurred. However, in terms of the latent variables, audience inclinations to racism, $\theta^m$ and $\theta^o$ were correlated more significantly, which further proves the existence of common underlying factors between modern and old-fashioned racism and suggests that the seeming dissonance between the two types of racism are a result of variance of contexts.

Other media provide a similar user experience as ABEMA, that allow audiences to post and read comments live while watching news programs. For example, YouTube Live and Niconico live (Japan), are used domestically or internationally. Our findings are applicable for understanding racial toxic commenting behavior on these media. Moreover, our findings on the association between news contents and audiences are important in these media than in ABEMA. The reason is that moderation over video content is loose in media, which allows users to upload user-generated-contents. Many opportunities can be exposed to biased and sensationalized video contents when watching them.

Although this study provided remarkable observations in the field of media and racism studies, it has some limitations. The effectiveness of the framework of this study should be verified with the datasets of other platforms regarding other minorities. In addition, if the dictionary-based approach of this expanded to include highly flexible methods, such as dictionary expansion by machine learning [71], it supports studies of prejudice against various minorities. We estimated news topics and their categories based on audiences' comments, instead of video data or metadata of news programs, which are difficult to acquire. Using such data would realize a more correct categorization of news topics.

## Conclusion

We investigated the associations between news contents and audiences in Japanese prejudice expressions against Koreans on the internet television. As a result, we found the following three clusters of such associations: 1) audiences who themselves are not so bigots occasionally express prejudice related to news contents (ambiguous clusters), 2) biased audiences express their prejudice, which seems indirectly relevant to news contents and have strong inducements to express racism (evocative), and 3) extremely biased audiences express their prejudice without any relevance to news contents, which have weak inducements to express racism (non-evocative). This finding provides a framework for understanding racial prejudice expressions with news contents.

## Materials and methods

### ABEMA

ABEMA is an online television service in Japan. It has approximately 20 channels that specialize in specific genres, such as news, sports, anime, and drama. Each channel broadcast programs with certain time schedules. The ABEMA news channels provide programs created by a Japanese TV broadcaster "TV Asahi"; thus, their topics and qualities are comparable to traditional television news. 6.38 million active ABEMA users per week, as recorded after the dataset used in this study were collected (Sep. 2018) [72].

Audiences of ABEMA can post 50-character comments while watching a program, and other users can read them immediately (Fig 1b). Users can hide the comment field; therefore, not all users read them. Most programs use the Japanese language and most comments are written in Japanese.

### Dataset

We used news channel logs recorded between Jan 17, 2018 and Aug 31, 2018. We divided the logs into 5 minutes grids to enable content-based analysis because news programs change subjects in a few minutes. This is not a theoretical, but a convenient basis of separation to ensure that each grid is sufficiently small in terms of the period and includes a sufficient number of comments. We extracted comments expressing modern and/or old-fashioned racism against Koreans using a dictionary-based classification criterion (Table 3), which were expansions of the criteria used in a previous study [36]. Those dictionaries were minor modifications of those used in a previous study [36] and fixed before the data collection in the present study. No retrospective adjustments to fit the present dataset were conducted.

This dictionary (Table 3) was deemed the expression of racism. Previous study showed that most of the tweets (approximately 90%) classified as racist with those criteria had negative

**Table 3. Extraction from word lists for detecting comments relevant to Koreans, modern racism, and old-fashioned racism (see S1 Dataset for complete lists of Japanese words).** We analyzed comments, including both words nominated in Korean-cues-lists and racism-cues-lists. Note that we removed comments that were irrelevant to Koreans using an exclusion word list (see S2 Dataset). For example, comments including "*Zainichi Beigun* (American Army stationed in Japan)" were excluded, although they include words in Korean-cues-lists.

| Label | Themes of Comments | Examples of Words in English |
|---|---|---|
| Korean | Korea and Korean | Korean, *Zainichi*, *Hwabyung* |
| Modern Racism | Rights of Koreans frequently noted as "privilege" | privilege, welfare, pension |
| Old-fashioned Racism | Inferiority of Koreans in morality, ability, and/or appearances | crime, vicious, rape |

impressions toward Koreans besides direct reference to racism-relevant themes [36]. In this study, we evaluated the validity of each dictionary (Table 3) as follows: 1) we made a list consist of 150 comments, the half of which refer to target racism and the other does not; for each racism (the classification were conducted based on one form of racism at once; thus the comments that referred only to non-target racism belonged to the latter half), two trained coders independently evaluated each comment whether they match the definition of relevant form of racism. The specific labels of racism were not presented to the coders. Instead, semantic definition of both forms of racism (e.g., "a blatant expression of the notion that Korean are inferior in some dimensions" for old-fashioned racism). As a result, inter-human consistencies were high enough and both dictionaries showed high consistencies with human evaluations (Table 4).

The racial toxic comment ratios of all comments were 0.011 (modern racism) and 0.019 (old-fashioned racism). These ratios included removed comments by the ABEMA moderators.

According to the inclusion of racist comments, audiences and grids included in this study was determined; audiences who expressed racial prejudice at least once and grids on which racism was expressed at least once were included.

Consequently, we obtained a dataset, in which $y_{ij}^x \in \{0, 1\}$ denotes that the audience $i$ did or did not express prejudice in the form of $x$ (modern/old-fashioned racism) at least once when $i$ was watching grid $j$ (this data is shown in S6 Dataset). S2 Table shows the frequencies and ratios of audiences, grids, and comments associated with the expression of modern or/and old-fashioned racism.

## Parameter estimation of statistical model

We estimated the parameters of Eqs 1 and 2 for modern and old-fashioned racism using Stan [67] (see S1 Source code); the number of chains was four, the number of iterations was 2,000, and the first 1,000 iterations were discarded as burn-in iterations. We used the following prior distributions: $a_j \sim \mathrm{N}(0, 10)$ (note that the domain of $a$ is $a \in [0, \infty)$), $b_j \sim \mathrm{N}(0, 1)$, $\theta_i \sim \mathrm{N}(0.1, 1)$, $\beta \sim \mathrm{N}(0, 1)$, where $N(\mu, \sigma)$ shows a normal distribution of mean $\mu$ and standard deviation $\sigma$. We used small standard deviations for the prior distributions of $b$, $\theta$, and $\beta$ to minimize their variance because they were multiplied by $a$, which has large standard deviations for the prior distributions of $a$. The standard deviation size of the prior distribution of $a$ (10) is large enough in terms of a logit function form on the statistical model (Eq 2). We used a biased means of the prior distribution of $\theta$ (0.1) because $b$ and $\theta$ cannot be determined uniquely when the news grids have small audiences. The models for both modern and old-fashioned racism were converged because of the estimation results; the values of Rhat (the Gelman and Rubin's convergence diagnostic [73]) for all parameters were less than 1.05.

## Clustering news grids

We divided the news grids into clusters for modern and old-fashioned racism with reference to the presence or absence of expression of target racism and the estimated parameters. First,

**Table 4. The results of the validation of dictionary based extraction for racism.** Concordance rates between coders one and two were 0.92 (modern racism) and 0.92 (old-fashioned racism).

|  | Modern racism | | Old-fashioned racism | |
|---|---|---|---|---|
|  | Coder 1 | Coder 2 | Coder 1 | Coder 2 |
| Recall | 0.94 | 0.92 | 0.88 | 0.89 |
| Precision | 0.89 | 0.93 | 0.87 | 0.91 |
| F-measure | 0.91 | 0.93 | 0.88 | 0.90 |

we labeled the clusters of grids on which target racism was not expressed as *non-expression* clusters. Then, we divided other grids where target racism was expressed at least once. S1 Fig shows the density of the distribution of $\hat{a}$s and $\hat{b}$s of all grids, except for those that belong to the non-expression cluster, for each type of racism. For both types of racism, the distributions had three peaks. Thus, the thresholds of $\hat{a}$ and $\hat{b}$ for each type of racism were manually established according to these visualized distributions. We labeled the clusters with small $a$ ($\hat{a}^m \leq 5.5$ and $\hat{a}^o \leq 7.0$) as *ambiguous* clusters because they had small discrimination around the evocation of content (i.e., $b$) and audience inclination (i.e., $\theta$). Then, we divided the remaining grids with large $a$ ($\hat{a}^m > 5.5$ and $\hat{a}^o > 7.0$) into two clusters for each type of racism. We labeled the clusters with large $b$ ($\hat{b}^m \geq 0.4$ and $\hat{b}^o \geq 0.5$) as *evocative* clusters, and those with small $b$ ($\hat{b}^m < 0.4$, and $\hat{b}^o < 0.5$) as *non-evocative* clusters, with respect to the content-dependent evocation of racism expression.

## News topic categories

To analyze the association between news topics and prejudice expression, the news grids were divided into affordable categories considering the covered contents. The logs used in this study excluded details of the contents; thus, we performed this categorization considering the resemblance of comments posted on each grid using topic modeling (latent Dirichlet allocation; LDA [74]).

The input data for LDA were created by the following procedure. First, up to 2000 comments posted on each grid were compiled into single text data. All comments were included in the data when a grid had less than or equal to 2000 comments; when more, 2000 comments were randomly extracted. Second, morphological analysis was conducted with the Japanese morphological analyzer MeCab [75] (we employed the mecab-ipadic-NEologd dictionary (https://github.com/neologd/mecab-ipadic-neologd) as the MeCab dictionary), and the potential indicators of news content (i.e., nouns) were extracted automatically. Third, stopwords (the words in [76]'s stopwords list and frequent but redundant words, such as pronouns mistakenly extracted as nouns (see S3 Dataset)) and words that did not appear frequently (less than 20 times across all data) were removed. Consequently, frequency vectors were prepared for the nouns that frequently appeared in each grid.

We performed LDA for the above data with parameters $K = 100$ and $\alpha = 50/100$ (number of topics and Dirichlet hyperparameter for topic proportions, respectively) and determined a list of news topics and their indicators. Then, we manually interpreted these topics considering the word frequencies and pointwise mutual information (PMI).

We implemented the categorization method used in previous study [77] and applied the following topic labels: political news, international news, social issues news, disaster news, crime news, nuclear accident news, incident news, entertainment news, sports news, political scandal news, celebrity/sports scandal news, lifestyle news, and weather news, in the following procedures along with [78]: 1) Two researchers independently applied these categories to the news topics, and 2) They decided the topic categories for the news topics by discussing intra-topic and inter-topic semantic validity. This list excludes topics irrelevant to specific news coverage, such as the introduction of cast members and reflections of audience attitudes (e.g., criticism of mass media, dissatisfaction with the Japanese government, and revilement to the Koreans). S4 Dataset listed the news topic categories, news topics, and words typical for each category (i.e., high-frequency and high-PMI words). The occurrence probabilities of combinations of news grids and topics are shown in S5 Dataset.

PMI in this procedure was defined as follows:

$$\text{PMI}_t(w) = \log_2 P(w|t) - \log_2 P(w) \tag{3}$$

where $t$ is a topic and $w$ is a word. $\text{PMI}_t(w) > 0$ denotes the word $w$ characteristic under the given condition of topic $t$.

### Ethics statement

The ABEMA users accept the terms of use and privacy policy, which allow the statistical analysis of their behavior data for academic research. ABEMA's privacy policy is available on the following site: https://abema.tv/about/privacy-policy (see the item 5). Commitments to protect personal information of the parent company of ABEMA, CyberAgent, Inc., are available on the following site: https://www.cyberagent.co.jp/en/way/security/privacy/ (see the section "purposes for collecting personal information").

## Supporting information

**S1 Table. Number of audiences/news grids/comments of each cluster of modern/old-fashioned racism.** The total number of all comments mentioning Koreans was 81,114.
(PDF)

**S2 Table. Frequencies and ratios of audiences, grids, and comments associated to modern/old-fashioned racism.**
(PDF)

**S3 Table. Co-occurrences of news grid clusters between modern/old-fashioned racism.** Columns and rows show modern/old-fashioned racism, respectively. The interclass correlation between both racism clusters was negative (Cohen's $\kappa = -0.332$ where $p$-value was less than $10^{-16}$). Diagonal elements were also relatively small. The news grids where audiences did not express any racism were not included in the dataset of this study, i.e., the cluster size of non-expression of modern and old-fashioned racism was zero.
(PDF)

**S1 Fig. Density distributions of a and b of modern/old-fashioned racism excluding non-expression clusters, and threshold lines for clustering news grids.**
(EPS)

**S1 Dataset. Word list of racism.**
(XLSX)

**S2 Dataset. Exclusion words for racism.**
(RTF)

**S3 Dataset. Additional stop words list.**
(TXT)

**S4 Dataset. Topic meanings and typical words.**
(XLSX)

**S5 Dataset. Occurrence probability matrix of news grids and topics.**
(CSV)

**S6 Dataset. Behavioral log data.** This shows whether audiences expressed their modern/old-fashioned racism when they watched news grids. The data structure is as follows: audience (audience ID), content (news grid ID), modern_racism (if an audience expressed modern

racism at a news grid then it is 1, else 0), and old-fashioned_racism (if an audience expressed old-fashioned racism at a news grid then it is 1, else 0), where audience IDs are pseudonymized and news grid IDs are numbered at random.
(CSV)

**S1 Source code. Stan code for our statistical model (Eqs 1 and 2).**
(STAN)

## Author Contributions

**Conceptualization:** Fumiaki Taka.

**Data curation:** Masanori Takano.

**Formal analysis:** Masanori Takano.

**Methodology:** Masanori Takano.

**Project administration:** Masanori Takano.

**Supervision:** Masanori Takano, Fumiaki Taka.

**Visualization:** Masanori Takano.

**Writing – original draft:** Masanori Takano, Fumiaki Taka.

**Writing – review & editing:** Fumiaki Taka, Soichiro Morishita, Tomosato Nishi, Yuki Ogawa.

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
