## [Decision Letter · Decision Letter 0]

2 Feb 2021

PONE-D-20-34442

Three clusters of content-audience associations in expression of racial prejudice while consuming online television news

PLOS ONE

Dear Dr. Takano,

Thank you for submitting your manuscript to PLOS ONE. After careful consideration, we feel that it has merit but does not fully meet PLOS ONE’s publication criteria as it currently stands. Therefore, we invite you to submit a revised version of the manuscript that addresses the points raised during the review process.

I am writing to let you know that your Manuscript ID JOU-20-0103.R2 entitled "Applying News Values Theory to Liking, Commenting, and Sharing Mainstream News Articles on Facebook" which you submitted to Plos One has now been reviewed.  The comments of the reviewers are included at the bottom of this letter.

Please note that the reviewers have recommended major revisions to your manuscript, which I believe will result in an improved manuscript overall. Therefore, I invite you to revise your manuscript, in conjunction with the comments of the reviewers.

We look forward to receiving your revised manuscript.

Kind regards,

Chang Sup Park, Ph.D.

Academic Editor

PLOS ONE

Journal Requirements:

2. Please include additional information regarding your study materials; in particular, please include a copy in English of the documents (S1 and S2) included as Supporting Information.

3.Please note that according to our submission guidelines (http://journals.plos.org/plosone/s/submission-guidelines), outmoded terms and potentially stigmatizing labels should be changed to more current, acceptable terminology. To that effect, please consider whether any terms used there or in your study might be considered offensive and confirm in the submission form or cover letter if that is not the case.

4. Please specify in your ethics statement in the Methods section and the submission form whether data were collected and analyzed anonymously.

5. In your Methods section, please include additional information about your dataset and ensure that you have included a statement specifying whether the collection method complied with the terms and conditions for the websites from which you have collected data.

6.Thank you for stating the following in the Competing Interests:

"MT and SM are employees of CyberAgent, Inc. There are no patents, products in development, or marketed products to declare."

We note that one or more of the authors have an affiliation to the commercial funders of this research study : CyberAgent, Inc.

Reviewers' comments:

Reviewer's Responses to Questions

**Comments to the Author**

1. Is the manuscript technically sound, and do the data support the conclusions?

Reviewer #1: Yes

Reviewer #2: No

2. Has the statistical analysis been performed appropriately and rigorously? 

Reviewer #1: Yes

Reviewer #2: I Don't Know

3. Have the authors made all data underlying the findings in their manuscript fully available?

Reviewer #1: Yes

Reviewer #2: Yes

4. Is the manuscript presented in an intelligible fashion and written in standard English?

Reviewer #1: Yes

Reviewer #2: Yes

5. Review Comments to the Author

Reviewer #1: This study tackles digital racism through an examination of racist discourses of Japanese

television viewers on ABEMA. The paper provides us with an important gap in our understanding of global television and news consumption and their implication for othering and racism. ABEMA case study is very good and innovative, I quite enjoyed reading their work. In order for the article to be considered for publication, it will have to be revised based on the below suggestions.

Major comments for the revision:

1) The authors should define the concept of racism. This can be done by appealing to critical race studies. Old fashioned and modern racism are not widely used in the existing literature. It is rather “new cultural racism” and “bio-racism”. Among many others I recommend the authors to conceptualise new cultural racism and bio-racism from the beginning of the paper using the works of Barker (1981), Shohat and Stam (2014) and/or Balibar (1991).

2) In this perspective, racialisation and racial formation could be good concepts to introduce to the paper. I recommend Omi and Winant, Bonilla-Silva and Beck et al. to define these concepts. The paragraph on the history of racism on the 2nd page is a good start but not sufficient. More historical, social and political context needed for readers who do not know the background of the historical tensions between Japan and Korea. Race and racism should be clarified in Japan in relation to its imperialism. “The antipathy toward Koreans by the Japanese can be partly attributed to extreme tension between the two Korean states and Japan over historical and territorial issues” – This really is not helpful to locate racism towards the Korean, please refrain from simplistic approaches.

3) On the second page the authors also discuss global racisms. This part should definitely include anti-refugee racism and Romaphobia because these two forms of racism are widely shared across specific national contexts and they are now widely accepted examples/forms of acceptable forms of racisms on a global scale.

4) The authors should define the concepts of audience behaviour and hate-speech as these are quite used but generally misunderstood terms. Especially the authors should bring in literature and discussions from audience research in relation to news consumption.

5) Several other studies included discussions on anonymity and its relation to online racism such as Keum and Miller 2018, Criss, Michaels and Solomon.. (2020) and Ozduzen, Korkut and Ozduzen 2020. The authors could benefit from a further contextualisation of online racism and how it is related to being anonymous.

6) “The majority of comments posted on these programs promoted an extremist right group and claims on the inferiority of ethnic Koreans in terms of morals, abilities, and appearances” – this looks quite like bio-racism not “new cultural racism”.

7) In terms of labelling the clusters as non-expression, ambiguous, evocative, and non-evocative, the authors need further explanation why they are named as such and what this naming entails.

Organisational comments:

1) The main objective of the study/its case study is introduced a bit late (55-56th lines on the 2nd page). It should be obvious from the start that the authors aim to unpick racism against Koreans on an online platform.

2) Same with the “material and methods” section – it should come way before than page 12.

3) There should be a conclusion section which wraps up all arguments and findings.

4) On top of the third page, the paragraph starting with “media exposure” could be connected to the previous paragraph in a better way.

5) Authors may wish to remove phrases like “to the best of our knowledge” and I also recommend the authors to read the piece once more even if the language use is good.

Reviewer #2: This manuscript addresses a relevant and interesting research topic, namely it explores the short term effects of television news on the expression of racial prejudice using the case of a Japanese online television provider. While the manuscript generally follows an intriguing idea and has interesting data to investigate, I have quite a few unanswered questions and methodological concerns that left me puzzled after reading it. I want to share my concerns below:

MAJOR: The authors explain that to the best of their knowledge, there is no study that investigates the short term effects of television news on the expression of racial prejudice. While this might be so, the authors do not explain why this may be important and different from what we know from studies about news on the internet or on social media. This is particularly relevant because the medium they study “ABEMA” seems to be a very specific and unique hybrid that merges television and social media aspects. What would we expect to be different?

MAJOR: While the data from “ABEMA” seems very useful for this study, I have to wonder how generalizable the findings are. Are there any similar media in other countries? I am particularly surprised by the non-existent content moderation that leaves me wondering about the journalistic quality of the platform and the non-representativity of the audiences using it.

MAJOR: Different forms of racism have been identified based on a dictionary approach. However, neither the validity of these dictionaries, nor the validation procedures used are being discussed in the manuscript. Automated approaches to text analysis are only as good as the human validation they are based on. Right now, this remains a complete mystery. See https://doi.org/10.1080/10584609.2020.1723752

MAJOR: I have similar concerns with the topic modelling. Was there any cross-researcher validation of the topics that resulted of the topic models? On ensuring the validity of topic models, see https://www.tandfonline.com/doi/full/10.1080/19312458.2018.1430754

MAJOR: I may have misunderstood the authors, but are they using the comments to identify the topic of the news casts? If so, this needs to be made clearer. Furthermore, I am wondering whether this measure can really be used to identify the topic of the news cast itself. I would at least want to see some clear evidence that comments and news casts are always clearly connected.

MINOR: I am missing a clear definition of the different forms of racism.

To sum up, I have quite a few concerns and open questions that need addressed appropriately before publication.

6. PLOS authors have the option to publish the peer review history of their article (what does this mean?). If published, this will include your full peer review and any attached files.

Reviewer #1: No

Reviewer #2: No

---

## [Author Response · Author response to Decision Letter 0]

13 Apr 2021

The corresponding reviewers' comments of the suggested changes are shown in the right margins of the manuscript, e.g., "R1-2" means Reviewer 1's comment 2.

[Response for the First Reviewer]

We would like to express our appreciation to the reviewer for his or her insightful comments, which have helped us significantly to improve the article.

- Comment 0

This study tackles digital racism through an examination of racist discourses of Japanese television viewers on ABEMA. The paper provides us with an important gap in our understanding of global television and news consumption and their implication for othering and racism. ABEMA case study is very good and innovative, I quite enjoyed reading their work. In order for the article to be considered for publication, it will have to be revised based on the below suggestions.

- Response 0

Thank you for your insightful comments. We have responded to all your comments and revised our manuscript as required.

[Major comments for the revision]

- Comment 1

The authors should define the concept of racism. This can be done by appealing to critical race studies. Old fashioned and modern racism are not widely used in the existing literature. It is rather “new cultural racism” and “bio-racism”. Among many others I recommend the authors to conceptualise new cultural racism and bio-racism from the beginning of the paper using the works of Barker (1981), Shohat and Stam (2014) and/or Balibar (1991).

- Response 1

Thank you for pointing out that there exists other useful conceptualizations different from McConahay's [1] and Sears's [2]. We fully recognize the usefulness of the concept of biological and cultural racism. However, McConahay's conceptualization also is often cited in social and political psychological field. Moreover, we acknowledged some advantages of McConahay's conceptualization in this study, thus continued to depend on it in the revised manuscript. However, we did not have any intentions to deny the usefulness and contributions of other conceptualizations. 

To make it clear, we have added in the Introduction (p.3, L.68) that some concepts resemble old-fashioned and modern racism (symbolic racism) as follows.

It is worth noting that there exist other conceptualizations of today's racism besides old-fashioned and modern racism (or symbolic racism). For example, biological and cultural racism [3] or blatant and subtle racism [4] are also frequently used instead of old-fashioned and modern racism. Among them, in this article, we depend on McConahay's terminology [1] for the following two reasons: it was technically difficult to automatically detect beliefs on biological and cultural casualties from the posted comments, thus the terms that refer to these facets are misleading; and, it has advantages in consistency and convenience to adopt the same conceptualization as the previous studies on racism against Koreans [5,6], including the availability of a well-developed dictionary (shown later).

However, we do not have intention to exclude conceptualizations other than McConahay's.

Additionally, we have described the definitions of racial prejudice and discrimination as follows to the Introduction (p.2, L.43).

Racism refers to prejudice and discrimination due to race or ethnic memberships. Prejudice, in turn, is defined as "the set of affective reactions, we have toward people as a function of their category memberships [7]." Affective reactions are based on certain beliefs on traits or behaviors of the group (i.e., stereotypes). Discrimination refers to behavioral bias and implies inappropriateness and unfairness [8].

- Comment 2

In this perspective, racialization and racial formation could be good concepts to introduce to the paper. I recommend Omi and Winant, Bonilla-Silva and Beck et al. to define these concepts. The paragraph on the history of racism on the 2nd page is a good start but not sufficient. More historical, social and political context needed for readers who do not know the background of the historical tensions between Japan and Korea. Race and racism should be clarified in Japan in relation to its imperialism. “The antipathy toward Koreans by the Japanese can be partly attributed to extreme tension between the two Korean states and Japan over historical and territorial issues” – This really is not helpful to locate racism towards the Korean, please refrain from simplistic approaches.

- Response 2

We agree that the description of the background in the previous version was insufficient for readers outside Japan. Thus, we added a paragraph to describe the historical background, which shortly but expounds on the role of modernization and imperialism in the specific case of racialization in Japan as follows (the Introduction; p.5, L.164) .

Racialization of Asian people can be traced back to Meiji period (A.D.1868-1921) in Japan [9]. To maintain collective self-esteem while confronting well-developed western cultures, Japanese and government required myths that Japanese ethnicity is superior to other Asian people. During invasion and colonial rule over Asia, the notion of racial hierarchy, which put the Japanese on the top became widely accepted. After the end of World War II, Japan waived dominance abroad, but derogation to Asian people continued [9]. In 1959, Japanese students sample rated Koreans most unfavorable among 12 race/ethnic groups [10]. Recently, some factors aggravate racial prejudice against Koreans by Japanese people. For example, the popularization of the internet and anonymous cultures [6], the spread of historical revisionism, which affirms Japan Empire and deny the existence or responsibility of Japanese' war crimes [6]. Additionally, the undergoing geopolitical conflicts between Japan and South Korea and between Japan and North Korea also contribute to the racial tension [11].

On the other hand, we thought that explanation of the concept of rationalization and racial formation in general is somewhat out of our scope, thus we did not include them.

- Comment 3

On the second page the authors also discuss global racisms. This part should definitely include anti-refugee racism and Romaphobia because these two forms of racism are widely shared across specific national contexts and they are now widely accepted examples/forms of acceptable forms of racisms on a global scale.

- Response 3

We agree that anti-refugee racism and Romaphoboa are a typical example of current racism. Thus, we included them in the paragraph in the Introduction (p.3, L.86) that gives examples of online racism to enhance readers' understanding.

- Comment 4

The authors should define the concepts of audience behaviour and hate-speech as these are quite used but generally misunderstood terms. Especially the authors should bring in literature and discussions from audience research in relation to news consumption.

- Response 4

We agree that some terms in our article should be clearly defined.

In this study, we analyzed audiences' watched contents, watching duration, and comment posting logs as audience behavior. We have added the details of audience behavior to the Introduction (p.4, L.132).

In this paper, we defined hate-speech as "abusive speech targeting specific group characteristics, such as ethnic origin [12]", although there are many other definitions which stress the importance of the purpose, inner motives, or ulterior consequences of the speech. This is because simple definition has an advantage because it is technically difficult to automatically detect such factors from posted comments. We have added this explanation to the Introduction (p.2, L.48).

For audience research, we had reviewed television news effects and political attitude of audiences (p.2, L14), and news compositions and prejudice expressions including hate speech (p.3, L.100). Thus, we did not add any descriptions in this article.

- Comment 5

Several other studies included discussions on anonymity and its relation to online racism such as Keum and Miller 2018, Criss, Michaels and Solomon.. (2020) and Ozduzen, Korkut and Ozduzen 2020. The authors could benefit from a further contextualisation of online racism and how it is related to being anonymous.

- Response 5

In response to this comment, we have discussed the relationships between anonymity and online racism in the Introduction (p.3, L.89) as follows.

Other researchers have stressed that the lack of social physical cues in online communication tempt users to accord to stereotypes and group norms, including about race [13]. The omnipresence of opportunities to form communities with people who share prejudicial attitudes are another factor [13], that would facilitate echo chambers [14].

- Comment 6

“The majority of comments posted on these programs promoted an extremist right group and claims on the inferiority of ethnic Koreans in terms of morals, abilities, and appearances” – this looks quite like bio-racism not “new cultural racism”.

- Response 6

We apologize that it was unclear that the sentence in question had mentioned both forms of racism in the previous version. To make it clear, we complemented the explanation about the extremist group's claiming in the Discussion (p.11, L.413).

Most of comments posted on these programs were propagation of an extremist right group's claiming existence of Zainichi Koreans' privilege for modern racism, and claims on the inferiority of ethnic Koreans in terms of morals, abilities, and appearances for old-fashioned racism.

- Comment 7

In terms of labelling the clusters as non-expression, ambiguous, evocative, and non-evocative, the authors need further explanation why they are named as such and what this naming entails.

- Response 7

In response to this comment, we have added explanations for labelling the three clusters in the caption of Table 1 (p.5).

[Organisational comments]

- Comment A

The main objective of the study/its case study is introduced a bit late (55-56th lines on the 2nd page). It should be obvious from the start that the authors aim to unpick racism against Koreans on an online platform.

- Response A

In response to this comment, we have included our main objective in the first paragraph of the Introduction (p.2, L.11).

- Comment B

Same with the “material and methods” section – it should come way before than page 12.

- Response B

Thank you for this important comment. We recognize that the order of sections you suggested is preferred in many fields, especially in social science (one of the authors of this article specialize in social psychology). However, in nature science and information science, articles are often structured in the order that is in this article. In fact, this journal's requirement is that authors present middle sections (materials and methods, results, discussion, and conclusions (optional)) in any order. See PLOS ONE's submission guideline "https://journals.plos.org/plosone/s/submission-guidelines#loc-manuscript-organization" for details.

For example, the following papers presented middle sections in the same order of our manuscript.

- Antonioni A, Tomassini M (2011) Network Fluctuations Hinder Cooperation in Evolutionary Games. PLOS ONE 6(10): e25555. {https://doi.org/10.1371/journal.pone.0025555}

- Verma P, Sengupta S (2015) Bribe and Punishment: An Evolutionary Game-Theoretic Analysis of Bribery. PLOS ONE 10(7): e0133441. {https://doi.org/10.1371/journal.pone.0133441}

- Ward ME, McMahon G, St Pourcain B, Evans DM, Rietveld CA, et al. (2014) Genetic Variation Associated with Differential Educational Attainment in Adults Has Anticipated Associations with School Performance in Children. PLOS ONE 9(7): e100248. {https://doi.org/10.1371/journal.pone.0100248}

Thus, we did not alter the order of the sections. 

Sorry for the inconvenience caused by difference in customs.

- Comment C

There should be a conclusion section which wraps up all arguments and findings.

- Response C

In response to this comment, we have added the conclusion section (p.14, L.537).

- Comment D

On top of the third page, the paragraph starting with “media exposure” could be connected to the previous paragraph in a better way.

- Response D

In response to this comment, we have revised the first part of the paragraph for better connection with the previous paragraph (p.3, L.110).

- Comment E

Authors may wish to remove phrases like “to the best of our knowledge” and I also recommend the authors to read the piece once more even if the language use is good.

- Response E

In response to this comment, we have removed the phrase in question (p.2, L.21). In addition, we have sent the manuscript for proofreading conducted by a native English speaker who is acquainted with academic writing.

[Response for the Second Reviewer]

- Comment 0

This manuscript addresses a relevant and interesting research topic, namely it explores the short term effects of television news on the expression of racial prejudice using the case of a Japanese online television provider. While the manuscript generally follows an intriguing idea and has interesting data to investigate, I have quite a few unanswered questions and methodological concerns that left me puzzled after reading it. I want to share my concerns below:

- Response 0

Thank you for this insightful comment. We have responded to all your comments and revised our manuscript as requested.

- Comment 1

MAJOR: The authors explain that to the best of their knowledge, there is no study that investigates the short term effects of television news on the expression of racial prejudice. While this might be so, the authors do not explain why this may be important and different from what we know from studies about news on the internet or on social media. This is particularly relevant because the medium they study “ABEMA” seems to be a very specific and unique hybrid that merges television and social media aspects. What would we expect to be different?

- Response 1

We agree with this comment. In response to this comment, we have added the following discussion in the Introduction (p.2, L.28). Additionally, we have fine-tuned the first sentence in the next paragraph.

The short-term effects of news broadcast on television or similar media (e.g., internet television), which many people consume at present, should also be investigated, because these news media provide experiences that differ from those with online news investigated in previous studies [15,16]. In contrast to online news sites where users can read each news content at their own pace and when it is convenient for them, television-like media requires them to watch it at fixed pace during scheduled time. Many audiences watching the same program simultaneously, and post and read comments live. With the passage of time, the contexts those comments are generated quickly become invisible for general audiences. In other words, comments on internet television are volatile, while other forms of online news have accumulated comments field. With these characteristics, television-like media affect the audience differently, such as imposing a time-pressures and enhancing thoughtless commenting behavior.

Thus, news transmitted via television-like media can have different short-term effects on audiences in different ways from other forms of online media.

- Comment 2

MAJOR: While the data from “ABEMA” seems very useful for this study, I have to wonder how generalizable the findings are. Are there any similar media in other countries? I am particularly surprised by the non-existent content moderation that leaves me wondering about the journalistic quality of the platform and the non-representativity of the audiences using it.

- Response 2}

In response to this comment, we have added more detail to the Discussion section (p.13, L.519).

The user experience of the online television, ABEMA, for real-time commenting to news videos is similar to YouTube Live and Niconico Live (Japan), which are used domestically or internationally. Our findings are applicable for understanding racial toxic commenting behavior on these media. Moreover, our findings about association between news contents and audiences could be more important in these media than in ABEMA. The reason is that moderation over video content is loose in media, which allow users to upload user-generated-contents. Many opportunities can be exposed to biased and sensationalized video contents when watching them.

ABEMA has been striving to remove racial toxic comments via natural language processing as with other news sites. In this study, we use all comment data including these removed comments for analyzing racial toxic expressions. The racial toxic comments ratio of all comments was small (modern racism: 0.011; old-fashioned racism: 0.019). We have added this explanation to the Introduction (p.4, L.136 and the footnote-2) and the Materials and Methods (p.14, L.580).

- Comment 3

MAJOR: Different forms of racism have been identified based on a dictionary approach. However, neither the validity of these dictionaries, nor the validation procedures used are being discussed in the manuscript. Automated approaches to text analysis are only as good as the human validation they are based on. Right now, this remains a complete mystery. See https://doi.org/10.1080/10584609.2020.1723752

- Response 3

We agree that the validity of our dictionaries is required for showing the applicability of dictionaries, as such we have conducted the validation of our dictionary-based extractions. The result is shown in Table 4 in the Materials and Methods (p.15).

We evaluated the validity of each racism dictionaries as follows: 1) we made the mixed lists of comments which matched each racism dictionary (Table 3) and comments which did not include such racism, 150 of each, and 2) two coders independently checked these comments whether based on the definitions of such racism. As a result, both racism dictionaries showed high correlations with human checks (Table 4).

We have added this procedure to the Materials and Methods (p.14, L.571).

- Comment 4

MAJOR: I have similar concerns with the topic modelling. Was there any cross-researcher validation of the topics that resulted of the topic models? On ensuring the validity of topic models, see https://www.tandfonline.com/doi/full/10.1080/19312458.2018.1430754

- Response 4

We agree that this validation process is required for our study, thus we have conducted cross-researcher validation of the topics that resulted in the topic models in the following procedures along with [17]: 1) two researchers independently applied these categories to the news topics, and 2) they decided topic categories for the news topics by discussing intra-topic and inter-topic semantic validity. We have added these procedures to the Materials and Methods (p.16 L.645).

Consequently, the four topic categories have been revised (see S4 Dataset Topic meanings and typical words). This change has had little effect on our analysis and discussion (the News topic categories of three clusters section).

- Comment 5

MAJOR: I may have misunderstood the authors, but are they using the comments to identify the topic of the news casts? If so, this needs to be made clearer. Furthermore, I am wondering whether this measure can really be used to identify the topic of the news cast itself. I would at least want to see some clear evidence that comments and news casts are always clearly connected.

- Response 5

As you inferred, we used audiences' comments to identify the topic of the news casts analyzing the relationships between the three clusters of racism expressions, racism type, and news topic categories (Table 2). Unfortunately, we could not validate such news topics and actual news topics because we did not have any data relating to the news topics, i.e., video files and metadata of news programs. We have described this limitation in the Discussion section (p.13, L.533).

- Comment 6

MINOR: I am missing a clear definition of the different forms of racism.

- Response 6

In response to this comment, we have added the explanation for modern and old-fashioned racism (p.3, L68).

References

[1] McConahay JB. Modern racism, ambivalence, and the Modern racism

scale. In: J F Dovidio, S L Gaertner, editors. Prejudice, discrimination,

and racism. San Diego, CA, US: Academic Press; 1986. p. pp. 91–125.

[2] Sears DO, Henry PJ. The origins of symbolic racism. Journal of personality

and social psychology. 2003;85(2):259–75. doi:10.1037/0022-3514.85.2.259.

[3] Kleinpenning G, Hagendoorn L. Forms of racism and the cumulative dimension of ethnic attitudes. Social Psychology Quarterly. 1993;56(1):21.

doi:10.2307/2786643.

[4] Pettigrew TF, Meertens RW. Subtle and blatant prejudice in western Europe. European Journal of Social Psychology. 1995;25(1):57–75.

doi:10.1002/ejsp.2420250106.

[5] Taka F. The anatomy of racism: Prejudice against Zainichi Koreans in the

age of the Iiternet. Tokyo: Keiso Shobo; 2015.

[6] Taka F. Quantitative and theoretical investigation of racism in Japan: A

social psychological approach. In: Higaki S, Nasu Y, editors. Hate speech

in Japan: The hate speech elimination act and its non-regulatory approach;

2020.

[7] David J Schneider. The psychology of stereotyping. New York: Guilford

Press; 2004.

[8] Dovidio JF, Hewstone M, Glick P, Esses VM. Prejudice, stereotyping and

discrimination: Theoretical and empirical overview. In: J F Dovidio, M

Hewstone, P Glick, V M Esses, editors. The SAGE Handbook of Prejudice,

Stereotyping and Discrimination. London, UK: SAGE Publications Ltd;

2010. p. 3–28.

[9] Banzai T. Formation of racial or ethcic stereotypes and prejudice in modern

Japan. Tokyo: Taga Shuppan; 2005.

[10] Haratani T, Matsuyama Y, Minami Y. Study on stereotypes and preferences among Japanese students toward themselves and other national ethnic groups. The Japanese Journal of Educational Psychology. 1960;8(1):1–7.

doi:10.5926/jjep1953.8.1 1.

[11] Higuchi N. Japan’s ultraright: Zaitokukai, foreigner’s suffrage and East

Asian geopolitics. Nagoya, Japan: The University of Nagoya Press; 2014.

[12] Warner W, Hirschberg J. Detecting hate speech on the world wide web.

In: Proceedings of the 2012 Workshop on Language in Social Media (LSM

2012); 2012. p. 19–26.

[13] Keum BTH, Miller MJ. Racism on the internet: Conceptualization and

recommendations for research. Psychology of Violence. 2018;8(6):782–791.

doi:10.1037/vio0000201.

[14] Criss S, Michaels EK, Solomon K, Allen AM, Nguyen TT. Twitter fingers

and echo chambers: Exploring expressions and experiences of online racism

using twitter. Journal of Racial and Ethnic Health Disparities. 2020; p. 1–

10. doi:10.1007/s40615-020-00894-5.

[15] Eveland WP, Seo M, Marton K. Learning from the news in

campaign 2000: An experimental comparison of TV news, newspapers, and online news. Media Psychology. 2002;4(4):353–378.

doi:10.1207/S1532785XMEP0404 03.

[16] Rasha A Abdulla, Bruce Garrison, Michael B Salwen, Paul D Driscoll,

Denise Casey. Online news credibility. In: Salwen MB, Garrison B, Driscoll

PD, editors. Online News and the Public. 1st ed. Lawrence Erlbaum Associates; 2004. p. 147–163.

[17] Maier D, Waldherr A, Miltner P, Wiedemann G, Niekler A, Keinert A,

et al. Applying LDA topic modeling in communication research: Toward

a valid and reliable methodology. Communication Methods and Measures.

2018;12(2-3):93–118. doi:10.1080/19312458.2018.1430754

---

## [Decision Letter · Decision Letter 1]

9 Jun 2021

PONE-D-20-34442R1

Three clusters of content-audience associations in expression of racial prejudice while consuming online television news

PLOS ONE

Dear Dr. Takano,

Thank you for submitting your manuscript to PLOS ONE. After careful consideration, we feel that it has merit but does not fully meet PLOS ONE’s publication criteria as it currently stands. Therefore, we invite you to submit a revised version of the manuscript that addresses the points raised during the review process.

The reviewers believe most of their concerns have been addressed, and I concur. But there are a few lingering questions that need clarification. Particularly, one reviewer wants to see a much clear explanation about the validation of the dictionary. Another reviewer suggests that you conceptualize TV audiences using existing audience research literature. Meticulous proofreading is required.

I look forward to receiving your revision.

We look forward to receiving your revised manuscript.

Kind regards,

Chang Sup Park, Ph.D.

Academic Editor

PLOS ONE

Journal Requirements:

Additional Editor Comments (if provided):

Reviewers' comments:

Reviewer's Responses to Questions

**Comments to the Author**

1. If the authors have adequately addressed your comments raised in a previous round of review and you feel that this manuscript is now acceptable for publication, you may indicate that here to bypass the “Comments to the Author” section, enter your conflict of interest statement in the “Confidential to Editor” section, and submit your "Accept" recommendation.

Reviewer #1: All comments have been addressed

Reviewer #2: All comments have been addressed

2. Is the manuscript technically sound, and do the data support the conclusions?

Reviewer #1: Yes

Reviewer #2: Partly

3. Has the statistical analysis been performed appropriately and rigorously? 

Reviewer #1: I Don't Know

Reviewer #2: Yes

4. Have the authors made all data underlying the findings in their manuscript fully available?

Reviewer #1: Yes

Reviewer #2: Yes

5. Is the manuscript presented in an intelligible fashion and written in standard English?

Reviewer #1: Yes

Reviewer #2: Yes

6. Review Comments to the Author

Reviewer #1: I believe the authors did an excellent job of revising this paper. The concepts are defined better (such as racism and hate speech) and there is more context for Japan for international readers (more could be done here). The added table is very good, summarising the findings. The part where the authors define racism towards the end of the 15th and the top of the 16th pages is good, but it concentrates a bit too much on McConahay’s take on racism. I still think TV audiences should be conceptualised in the paper using existing audience research literature. Also, the paper would still benefit from a good read of its English: for example, imposing a time-pressures -15th page/Not romas but roma – 16th page. Also, I think this sentence should be removed: “However, we do not have intention to exclude conceptualizations other than McConahay’s”. Finally, on the 18th page, right before the title “results”, there are some fragmented very short paragraphs – could be good if the authors turn them into better a better flowing longer paragraph. Looking forward to reading the article’s published version.

Reviewer #2: I want to thank the editor and the authors for having given me the opportunity to review this manuscript a second time. I really believe that the authors tried to address all my previous comments. Generally, I do believe that the manuscript is now ready to publish.

However, I would like them to clarify a little better what they did in the validation of the dictionary first. Particularly part 2 of their validation seems a bit ominous: "2) two trained coders independently evaluated each comment whether they match the definition of relevant form of racism". When validating the dictionaries, did you test whether it the respective racism dictionary discriminates against non-racist comments or against comments including the second type of racism? Judging from the high validity measures, I fear that they did the former. So, we don't really know how well it discriminates between the two types of racism. If it actually is the latter, I am a bit skeptical that the authors may have over-fitted the dictionary to their specific data.

Otherwise all concerns have been addressed either by clarification or addressing the limitation in the text.

7. PLOS authors have the option to publish the peer review history of their article (what does this mean?). If published, this will include your full peer review and any attached files.

Reviewer #1: No

Reviewer #2: No

---

## [Author Response · Author response to Decision Letter 1]

29 Jun 2021

# Response for the First Reviewer

[Comment 1]

I believe the authors did an excellent job of revising this paper. The concepts are defined better (such as racism and hate speech) and there is more context for Japan for international readers (more could be done here). The added table is very good, summarising the findings. The part where the authors define racism towards the end of the 15th and the top of the 16th pages is good, but it concentrates a bit too much on McConahay’s take on racism. I still think TV audiences should be conceptualised in the paper using existing audience research literature. Also, the paper would still benefit from a good read of its English: for example, imposing a time-pressures -15th page/Not romas but roma – 16th page. Also, I think this sentence should be removed: “However, we do not have intention to exclude conceptualizations other than McConahay’s”. Finally, on the 18th page, right before the title “results”, there are some fragmented very short paragraphs – could be good if the authors turn them into better a better flowing longer paragraph. Looking forward to reading the article’s published version.

[Response 1]

Thank you for your positive evaluation of our revisions.

We have revised our manuscript based on your suggestion:

- We have described the relevance of this study to audience researches in the Introduction (p.2, L.42) and the Discussion (p.11, L.375) sections.

- We have carefully reviewed the manuscript and revised some expressions, including "a time-pressures" and "romas."

- According to your comment, we have removed the sentence "However, we do not have intention to exclude conceptualizations other than McConahay's."

- We have also changed the fragmented paragraphs right before the Results section in the following:

 - 1) We have removed the sentence "Previous studies have shown that the expression of both modern and old-fashioned racism is frequent in Japanese social media, as cited above [7,32]" because there was a similar sentence (the 12th paragraph) in the Introduction (L.103) section; 

 - 2) we have moved the sentence "The antipathy toward Koreans by the Japanese can be partly attributed to extreme 177 tension between the two Korean states and Japan over historical and territorial 178 issues [36,58,61,62]" to the last part of the 21th paragraph (L.170) in the Introduction section. 

# Response for the Second Reviewer

[Comment 1]

I want to thank the editor and the authors for having given me the opportunity to review this manuscript a second time. I really believe that the authors tried to address all my previous comments. Generally, I do believe that the manuscript is now ready to publish. However, I would like them to clarify a little better what they did in the validation of the dictionary first. Particularly part 2 of their validation seems a bit ominous: "2) two trained coders independently evaluated each comment whether they match the definition of relevant form of racism". When validating the dictionaries, did you test whether it the respective racism dictionary discriminates against non-racist comments or against comments including the second type of racism? Judging from the high validity measures, I fear that they did the former. So, we don't really know how well it discriminates between the two types of racism. If it actually is the latter, I am a bit skeptical that the authors may have over-fitted the dictionary to their specific data. Otherwise all concerns have been addressed either by clarification or addressing the limitation in the text.

[Response 1]

Thank you for your comments about the concerns over the validation process and dictionary over-fitting.

In the validation process, we classified the comments in view of one specific form of racism at once. In other words, the comments that did not refer to target racism but did to the other form of racism belonged to the contrast group among those did not refer to either form of racism. In addition, the dictionaries used in the present study were fixed before the data collection by minor revisioning of a previous study, and no retrospective adjustments to fit our specific dataset were conducted. Thus, the high accuracy measures is not a result of inadequate comprarisons or over-fitted dictionaries. To make this point clear, we have added the above description to the Dataset in the Materials and Methods (L.575 and L.582) section.

---

## [Decision Letter · Decision Letter 2]

12 Jul 2021

Three clusters of content-audience associations in expression of racial prejudice while consuming online television news

PONE-D-20-34442R2

Dear Dr. Takano,

We’re pleased to inform you that your manuscript has been judged scientifically suitable for publication and will be formally accepted for publication once it meets all outstanding technical requirements.

Kind regards,

Chang Sup Park, Ph.D.

Academic Editor

PLOS ONE

Additional Editor Comments (optional):

Reviewers' comments:

Reviewer's Responses to Questions

**Comments to the Author**

1. If the authors have adequately addressed your comments raised in a previous round of review and you feel that this manuscript is now acceptable for publication, you may indicate that here to bypass the “Comments to the Author” section, enter your conflict of interest statement in the “Confidential to Editor” section, and submit your "Accept" recommendation.

Reviewer #1: All comments have been addressed

Reviewer #2: All comments have been addressed

2. Is the manuscript technically sound, and do the data support the conclusions?

Reviewer #1: Yes

Reviewer #2: (No Response)

3. Has the statistical analysis been performed appropriately and rigorously? 

Reviewer #1: Yes

Reviewer #2: Yes

4. Have the authors made all data underlying the findings in their manuscript fully available?

Reviewer #1: Yes

Reviewer #2: Yes

5. Is the manuscript presented in an intelligible fashion and written in standard English?

Reviewer #1: Yes

Reviewer #2: Yes

6. Review Comments to the Author

Reviewer #1: Looking forward to reading this research published! Many thanks for all the authors to adequately and meticulously address all of my revision requests.

Reviewer #2: Thanks for these additions! I have no further questions. I am looking forward to see this published!

7. PLOS authors have the option to publish the peer review history of their article (what does this mean?). If published, this will include your full peer review and any attached files.

Reviewer #1: **Yes: **Dr Ozge Ozduzen

Reviewer #2: No

---

## [Editor Report · Acceptance letter]

14 Jul 2021

PONE-D-20-34442R2 

Three clusters of content-audience associations in expression of racial prejudice while consuming online television news 

Dear Dr. Takano:

I'm pleased to inform you that your manuscript has been deemed suitable for publication in PLOS ONE. Congratulations! Your manuscript is now with our production department. 

Kind regards, 

on behalf of

Dr. Chang Sup Park 

Academic Editor

PLOS ONE